# Prevalence, Population Diversity and Antimicrobial Resistance of *Campylobacter*
*coli* Isolated in Italian Swine at Slaughterhouse

**DOI:** 10.3390/microorganisms8020222

**Published:** 2020-02-07

**Authors:** Guido Di Donato, Francesca Marotta, Roberta Nuvoloni, Katiuscia Zilli, Diana Neri, Daria Di Sabatino, Paolo Calistri, Elisabetta Di Giannatale

**Affiliations:** 1Istituto Zooprofilattico Sperimentale dell’Abruzzo e del Molise “G. Caporale”, National Reference Centre for Veterinary Epidemiology, Programming, Information and Risk Analysis, 64100 Teramo, Italy.; 2Istituto Zooprofilattico Sperimentale dell’Abruzzo e del Molise “G. Caporale”, National Reference Laboratory for Campylobacter, 64100 Teramo Italy; k.zilli@izs.it (K.Z.); d.neri@izs.it (D.N.); e.digiannatale@izs.it (E.D.G.); 3Department of Veterinary Sciences, University of Pisa, Pisa, Italy; roberta.nuvoloni@unipi.it

**Keywords:** *Campylobacter coli*, molecular analysis, antimicrobial resistance

## Abstract

*Campylobacter* spp. are among the microorganisms most commonly associated with foodborne disease. Swine are known to be the main reservoir of *Campylobacter coli* and a possible source infection of humans as a result of carcass contamination at slaughter. The aim of this study was to evaluate the prevalence of *C. coli* contamination in swine carcasses, the antimicrobial resistance (AMR) patterns of isolates and the genetic diversity between strains obtained from swine and those isolated from humans. The prevalence of contamination was higher on carcasses (50.4%) than in faeces (32.9%). The 162 *C. coli* isolated from swine were examined by pulsed-field gel electrophoresis (PFGE) and multi-locus sequence typing (MLST). The results of PFGE indicated a high genetic diversity among the isolates, with 25 different PFGE types. MLST assigned 51 sequence types (STs) to isolates. The most common genotype was ST-854 (16.04%), ST-9264 (10.49 %) and ST-1016 (6.08 %). Results of AMR showed a high resistance to quinolones and fluoroquinolones together with aminoglycosides and tetracycline. Many strains were multi-resistant with predominant R-type TeSCipNa (57%). Five resistance genes were detected along with mutation in the gyrA gene. A strong correlation between phenotypic and genotypic resistance was found for fluoroquinolone and tetracycline. Genetic profiles obtained in swine isolates were compared to those of 11 human strains. All human strains and 64.19% of animal strains (104/162) were assigned to the ST-828 clonal complex.

## 1. Introduction

*Campylobacter* spp. are among the most common causes of bacterial diarrhoea worldwide and are estimated to cause approximately 246,000 illnesses annually in the European Union (EU), mostly due to consumption of contaminated food [1]. *Campylobacter* may be transferred to humans indirectly through the ingestion of contaminated water or food [2] and less frequently by direct contact with contaminated animals or animal carcasses. The species most commonly associated with human infection are *Campylobacter jejuni* followed by *Campylobacter coli* and *Campylobacter lari,* although other *Campylobacter* species, including the non-thermophilic *Campylobacter fetus*, are known to occasionally cause human infection [1].

*C. jejuni* is considered the most frequent *Campylobacter* species associated with disease in humans, and are responsible for about 80%–90% of the total number of human cases of campylobacteriosis in the EU [1]. However, different studies have highlighted the importance of *C. coli* as an emergent problem in public health due to its greater resistance to antibiotics [3,4]. In the EU, *C. coli* has been found to be responsible for about 9% of human campylobacteriosis in the EU. Food producing animals like poultry, cattle and swine are common hosts and important reservoirs of *Campylobacter* species. *C. jejuni* is considered prevalent in poultry [5] and cattle [6], while pigs are mostly implicated as reservoirs of *C. coli* [7]. 

Pigs are often sub-clinically infected with *Campylobacter* spp. and contamination of meat during meat processing remains an important food safety risk [3,4,8]. Previous studies estimated the prevalence of contamination in pigs varying between 50% and 100%, with excretion levels ranging from 102 to 107 Colony Forming Units (CFU) of *Campylobacter* per gram of faeces [9]. This study aimed at estimating the prevalence and levels of contamination of thermotolerant *Campylobacter* in faeces and carcasses of pigs during slaughtering. An evaluation of possible correlations between the genotypic and phenotypic expressions for resistance to antimicrobials in the isolated strains was performed. A comparison between the multi-locus sequence typing (MLST) profiles obtained from pig and human strains, isolated in the same time period, was also conducted.

## 2. Materials and Methods 

### 2.1. Study Design

The sampling was carried out in a pig slaughterhouse located in the Abruzzo region of Italy. The sampling activities were carried out during sixteen sessions along the whole year, with four sessions for each season. A total of 12,308 animals were slaughtered, among which 280 animals were randomly selected (1 carcass for every 40–50 animals, about 17–18 animals sampled for each visit). This sample size was calculated to be able to estimate the prevalence of contamination with 6% of precision, considering 50% of expected prevalence and 95% of confidence level [10]. For all the animals sampled, the fattening phase of pigs was carried out in Italy, in 18 farms (coded F1 to F18) located in different regions of north (Piemonte, F3, F16, F9 and Emilia Romagna F1, F7, F10), central (Umbria F8, F12 and Abruzzo F2, F4, F5, F6, F11, F13, F15) and south Italy (Puglia F14, F17, F18) Figure 1. From each animal, the faecal content was taken immediately after the evisceration phase while swab samples from carcasses were collected before cooling, with a sampled surface of 400 cm^2^ for each carcass (withdrawal points: ham, back, belly, jowl). All samples were transported at 4 °C in refrigerated boxes and processed immediately on return to the laboratory. 

#### 2.1.1. *Campylobacter* Culture and PCR Typing

Faecal samples were cultured in Preston broth (Biolife, Milan, Italy) and incubated under microaerophilic conditions at 41.5 °C for 24 h. After incubation, 100 microliters of pre-enrichment broth were plated in duplicate on mCCDA and Karmali plates and incubated under microaerobic conditions at 41.5 °C for 48 h. Isolation and enumeration of thermotolerant *Campylobacter* were performed, respectively, according to part 1 and to the part 2 of the EN ISO 10272-2006 on swab samples. The isolates identified as *Campylobacter* spp. were then submitted to species identification by a multiplex PCR method, as previously described [11]. DNA was extracted using the Maxwell 16 Tissue DNA Purification Kit (Promega Corp., Madison, WI, USA) according to the manufacturer’s instructions and quantified using a Nanodrop Spectrophotometer (Nanodrop Technologies, Celbio Srl., Milan, Italy).

### 2.2. Pulsed-Field Gel Electrophoresis (PFGE)

Pulsed-field gel electrophoresis (PFGE) was performed according to the instructions of the 2013 U.S. PulseNet protocol for *Campylobacter* [12]. *C. coli* strains were sub-cultured on Columbia agar at 41.5 °C for 48 h in microaerophilic atmosphere and embedded in agarose blocks (Seakem Gold agarose, Lonza, Rockland, NY, USA). The blocks were then lysed, washed and digested with SmaI and KpnI enzymes (Promega, Italy), 25U at 25 °C for 4 h and subjected to pulsed-field electrophoresis in 1% agarose gel (Seakem Gold agarose, Lonza) for 18 h (Chef Mapper XA, Biorad Laboratories, Hercules, CA, USA). *Salmonella* serovar Branderup H9812 digested with XbaI enzyme (Promega, Milan, Italy), was used as standard molecular weight size. The gel was stained with Sybr Safe DNA gel stain (Invitrogen) and photographed at transilluminator (Alpha Innotech). The image analysis was performed using the program Bionumerics v. 7.6 (Applied Maths NV, Sint-Martens-Latem, Belgium). Level of similarity was calculated with the Dice correlation coefficient (position tolerance was set at 1%), and the unweighted pair group mathematical average UPGMA clustering algorithm was used for cluster analysis of the PFGE pattern. PFGE-clusters were defined at 100% similarity between macrorestriction patterns [13]. Untypeable isolates were not included in the analysis.

### 2.3. Multi-Locus Sequence Typing (MLST)

MLST was performed as described by Dingle et al. [14] for all *C. coli* isolates. MLST amplifies a segment of 7 housekeeping genes: aspA (aspartase, 477 bp), glnA (glutamine synthase, 477 bp), gltA (citrate synthase, 402 bp), glyA (serine hydroxyl methyl transferase, 507 bp), pgm (phosphor glucomutase, 498 bp), and tkt (transketolase, 459 bp) and uncA (ATP synthase, alpha subunit, 489 bp), to yield a total composite sequence length (all 7 loci) of 3309 bp. Oligonucleotides primers for the PCR and cycle sequencing reactions were carried out according to the *Campylobacter* MLST website [15]. Briefly, purified PCR products were sequenced by using the ABI PRISM BigDye® Terminator 3.1 Cycle Sequencing Kit (Applied Biosystems) according to the manufacturer and analyzed with the ABI PRISM 3500 Genetic Analyzer (Applied Biosystems). The alleles, sequence types (STs) and clonal complexes (CCs) were identified using the MLST database available online [16]. Novel alleles were submitted to the PubMLST *C. jejuni/C. coli* databases curators for number assignment. A minimum spanning tree (MST) of the results was generated in PHILOVIZ 2.0 [17] using the goeBURST algorithm [18]. In this phylogenetic tree construction, the sequences of the seven house-keeping genes analyzed (including MLST allelic sequences and flanking regions) were aligned by MUSCLE in MEGA 5.0 [19]. A phylogenetic tree was built using the maximum parsimony method and was analysed by 1000 replicates in the bootstrap test.

### 2.4. Antimicrobial Susceptibility and Resistance Genes

Susceptibility to antimicrobials was evaluated with the microdilution method using the Sensititre automated system (TREK Diagnostic Systems, Italy) following the harmonised rules for the monitoring and reporting of AMR in Europe (Commission Implementing Decision 2013/652/EC). Colonies were cultured on Columbia agar for 48 h in microaerophilic atmosphere, inoculated in Mueller Hinton Broth supplemented with blood and dispensed into Eucamp microtiter plates (TREK Diagnostic Systems, Biomedical Service, Italy), containing known scalar concentrations of the following antimicrobial substances: gentamicin (GEN) (0.12–16 µg/mL), streptomycin (STR) (1–16 µg/mL), ciprofloxacin (CIP) (0.06–4 µg/mL), tetracycline (TET) (0.25–16 µg/mL), erythromycin (ERY) (0.5–32 µg/mL), nalidixic acid (NAL) (2–64 µg/mL), and chloramphenicol (CHL) (2–32 µg/mL). The plates were then incubated at 42 °C in microaerobic atmosphere for 24 h. To evaluate the MICs of the isolates, Swin v3.3 Software (Thermo Fisher Scientific) was used in accordance with the epidemiological cutoff values (ECOFFs) as defined by EUCAST (European Committee on antimicrobial breakpoints) (www.eucast.org) to interpret their antimicrobial susceptibilities. *C. jejuni* strain NCTC 11351 was included for the quality control of the minimal inhibitory concentration (MIC) test. Strains were considered resistant when MIC break points were ≥ to 0.5 ≥ to 8, ≥ to 2, ≥ to 16, ≥ to 4 and ≥ to 2, for respectively, ciprofloxaxin, erytromycin, gentamicin, nalidixid acid, streptomycin and tetracycline. *C. coli* genome assemblies, available at the NRL for *Campylobacter*, were searched for genomic AMR traits presence. AMR genes were identified in silico using ABRicate v. 0.8 (Available online: https://github.com/ tseemann/abricate/) and by querying the publicly available Comprehensive Antibiotic Resistance Database [20]. Assemblies were annotated using Prokka v1.13 [21] and *gyrA* sequences were extracted using the query_pan_genome function in Roary v3.12.0 [22]. *GyrA* genes were aligned using Uniprot UGENE v1.18.0 [23]. Only mutations in the quinolone resistance-determining region (QRDR) of *gyrA* were considered to be the determinants of resistance, being these loci linked with phenotypic resistance to quinolones. In detail, for *gyrA*, we analyzed the amino acid changes at position 86.

## 3. Results

A total of 280 pig carcasses were sampled at the slaughterhouse coming from different part of Italian Regions (Figure 1); the number of pigs finally sampled per farm was 10, 48, 7, 10, 10, 33, 12, 2, 1, 6, 2, 3, 6, 5, 1, 4, 1, 1 for farm F1, F2, F3, F4, F5, F6, F7, F,8, F9, F10, F11, F12, F13, F14, F15, F16, F17 and F18.

Among the 280 pigs, 162 females (10.7%) and 118 males (89.3%) and were represented by 216 fat (77.14 %) and 64 lean (22.86%). Almost all the animals (93.57%) were born in Italy, while 6.43% originated from other European countries but were fattened in Italy. *Campylobacter* has been isolated from both carcass swabs and faeces taken directly from intestinal loops of slaughtered animals. The prevalence of contamination was higher on carcasses (50.4%, 95% C.I.: 44.5%–56.2%) than in faeces (32.9%, 95% C.I.: 27.6%–38.6%). This difference was statistically significant (χ^2^McNemar 14.2, p = 0.002) during all the seasons. Only in summer, the prevalence of *C. coli* contamination was higher in faeces than in carcasses, as shown in Figure 2. *Campylobacter* contamination was low on carcasses: less than 10 cfu/cm^2^ in 90.8% of samples, between 10–50 cfu/cm^2^ and between 100-150 cfu/cm^2^ in 7.8% and 1.4% of samples, respectively. All positive animals were contaminated with *C. coli,* and three animals were also positive for *C. jejuni*. A total of 221 C. coli was isolated from swabs and faeces samples collected on pigs at slaughtering. A high genetic diversity for *C. coli* was observed, with both typing methods. PFGE profiles of *C. coli* strains, after SmaI and KpnI enzyme digestion, resulted in 25 clonal populations, according to a similarity of 90%. The most representative are shown in Table 1. In addition, during the same period, *C. coli* was isolated from 11 gastroenteritis hospitalized patients in the Abruzzo region. MLST analysis of 173 strains (162 from pigs and 11 from humans) of *C. coli* is shown in Figure 3. Almost all human isolates (90.90%) and 96/162 of swine isolates (64.19%) were assigned to the ST-828 complex. Among the pig isolates, the most frequent STs were ST-854 (16.04%), ST-9264 (10.49%) ST-1016 (6.8%) and ST-1108 (5.55%), as shown in Figure 3. Sequence types ST-828 (4.05%), ST-829 (2.31%), ST-1055 and ST-827 (1.73%) were present both in human and pig isolates, as shown in Figure 3. Each analyzed farm was characterized by different STs and nine of them by seven different PFGE pulsotypes (P1-P7), as shown in Table 1. In particular, farm 2, showed the presence of 10 different STs followed by farm 6 with 7 STs, as shown in Table 1. It is apparent that some ST profiles of *C. coli* appear to dominate in a geographic area for a variable period. For example, some genotypes, such as ST-1617, ST-9264, ST-1016, ST-828 and ST-1108, were isolated in subsequent seasons in animals coming from the same farms, as shown in Table 1. On the other hand, the same ST was also found on different farms and, with the exception of two of these cases (ST-9264 and 854-ST), the isolates had the same PFGE pulsotype. This is the case of ST-1016 predominant in five farms (F2-F3-F4-F5-F6) with PFGE pulsotype P2; ST-1617 and ST-1108, respectively, common in three farms (F1, F2, F6) and (F2, F6 and F10) with PFGE pulsotypes P1 and P7. Table 1. ST-9264 and ST-854 were found to be the most diverse ST with two different PFGE types. In many animals, the same ST was isolated in both the faeces and the carcass, as shown in Table 1. The comparison of the STs by country of origin of the animals showed the presence of the same STs in the majority of cases, although distinct STs between animals born in Italy and abroad were found in 27.6% of the pigs. 

The highest level of antimicrobial resistance of *C. coli* isolated from pigs was observed against quinolones and fluoroquinolones (74.66% of isolates for nalidixic acid and 70.13% for ciprofloxacin) together with aminoglycosides (90.95% of isolates for streptomycin) and tetracyclines (90.95% of isolates), as shown in Figure 4. Only 37.55% and 25.79% of the strains from pigs were resistant to Erytromycin and Gentamycin, respectively. *C. coli* strains isolated from humans showed similar antimicrobial resistance patterns: all isolates were resistant to quinolones and fluoroquinolones and 90.90% to tetracycline. Resistance to erythromycin was also quite high in both pigs (37.55%) and clinical isolates (20.0%). Streptomycin is revealed statistically more resistant in pig with respect to humans. Many strains were multi-resistant with a predominant R-type TeSCipNa (56.5%). The analysis of the isolates allowed to identify several resistance genes that include: gyrA, tet (O), cmeA, cmeB, cmeC and cmeR. The study of genotypic resistance by sequencing the QRDR region of the gyrA gene showed the presence of specific mutations responsible for quinolone and fluoroquinolone resistance in the genome of all human isolates and in 64.81% of the pig isolates. A different percentage was observed for bacterial strains containing tet (O) gene, identified in 72.72% and in 88.88% of human and pig isolates, respectively. The CmeABC multidrug efflux pump with its CmeR regulator gene was identified in every human and pig strain analysed, covering a range between 99.38% and 100%. Strong correlations between phenotypic and genotypic resistance were found for fluoroquinolones and tetracycline in pig and human isolates. In particular, we observed a high level of concordance for the two resistance rates for gyrA (92.41% and 100%) and tet (O) (97.70% and 80.8), in swine and clinical cases, respectively.

## 4. Discussion

Pigs are considered a possible source of *Campylobacter* infection of humans, and the high percentage of contaminated pig carcasses in this study confirm the role of this animal species as one of the main animal reservoirs of *C. coli* [24,25]. Furthermore, *C. coli* shows high levels of antimicrobial resistance, thus representing an additional public health problem. A high prevalence of *Campylobacter* in pigs is frequently observed in many countries: levels from 0% to 92.7% were reported for European members and no member states [24,26,27,28]. Differences in sampling and analytical methods may explain these results. In this study, all isolates from pigs were found to be *C. coli.* A co-infection with *C. jejuni* has only been observed in three animals.

This is in agreement with a previous report from Italy, showing low levels of *C. jejuni* colonization in pigs [29]. The higher prevalence of *Campylobacter* in carcasses compared to faeces could indicate a cross contamination during the slaughter process, argued by the low level of contamination and also confirmed by PFGE analysis. The higher percentage of positive faeces recorded during the warm seasons are consistent with the *Campylobacter* seasonality reported in pig [30]. *C. coli* is often present in the intestinal tracts of swine, and carcasses may be contaminated during evisceration, which is considered the most critical phase of slaughter, with possible cross-contamination between carcasses, if they have the possibility to be in contact along the slaughtering line or the equipment used for evisceration is not properly cleaned and disinfected between carcasses [29]. However, contamination levels are significantly reduced by the chilling of the carcasses [31] and *Campylobacter* is found only at low levels in pork retail products [32,33,34]. Despite not be able to grow out of the body of the living host, *Campylobacter* can survive in food products at refrigeration temperatures from one to three weeks. This clearly indicates the importance of the contamination of carcasses (and therefore of the meats), given the capacity of *Campylobacter* to survive after refrigeration and the following processing phases, and determining infections in the consumers [29]. Each farm from which the tested pigs originated resulted as characterized by different clonal populations, characterized by several STs. It is noteworthy that some of these STs, circulating in different farms and seasons, show a specific spatial and temporal distribution. In particular, ST-9264 was present in three different farms. In detail, in farm 2, this clone was isolated in winter in faeces and carcasses, demonstrating a contamination of the carcasses from the faeces; furthermore, it has also been isolated in the carcasses during the following season, therefore showing a temporal resistance. 

The ST-1016 828 complex was isolated in the same farms on carcasses of animals in two following seasons, while it was isolated in three other farms during one season, suggesting that this could be a strain adapted to pigs. ST-9291 was isolated in summer in one farm from faeces of animals reared in central Italy but born in HU, suggesting that this could be a strain introduced from abroad. There was no similarity between strains isolated from different lots, indicating that there was no persistence of strains in the slaughterhouse. These populations were prevalent in different seasons and have been isolated from animals born, raised and fattened on holdings in different Italian regions. The seasonality of the blocks and the link to the different farms seems to be confirmed for some strains. Several studies described the antimicrobial resistance of *Campylobacter* isolates from pigs [35]. The results of this study apparently showed higher percentages of antimicrobial resistance against ciprofloxacin, nalidixic acid, tetracycline and streptomycin, than those observed in other EU countries. In this study, *C. coli* strains resulted highly resistant to tetracyclines (90.95%), streptomycin (90.95%), and to quinolone (ciprofloxacin and nalidixic acid, 70.13% and 74.66 of the isolates, respectively). Macrolides, important compounds for the treatment of human campylobacteriosis, showed relevant levels of resistance (37.55%). In *C. jejuni, tet.*(O) was plasmid-encoded in 54% of tetracycline-resistant isolates, whereas in *C. coli,* tet(O) appeared to be located on the chromosome [35,36,37]. The major mechanism of tetracycline resistance in *Campylobacter* spp. is the binding and protection of ribosomal A site by the protein Tet(O). Ciprofloxacin resistance in *C. jejuni* and *C. coli* is mainly due to point mutations in the quinolone resistance-determining region (QRDR) of the GyrA protein [38]. Erythromycin resistance in *Campylobacter* spp. has been associated with mutations in the 23S rRNA and in the large loop of the L4 and L22 50S ribosomal proteins [39]. Multidrug efflux can also contribute to reducing the intracellular concentration of several antibiotics, including tetracycline, ciprofloxacin and erythromycin [38]. In the EU, the proportion of *C. coli* isolates resistant to both ciprofloxacin and erythromycin (used in human campylobacteriosis) is low (10.2%), although Portugal, Spain and Finland reported in 2017 high proportions of strains with combined resistance to these two antimicrobials. An increasing proportion of *C. coli* isolates resistant to ciprofloxacin and tetracyclines was observed in many countries from 2014 to 2017. Thirteen EU countries reported in 2017 a proportion of ciprofloxacin-resistant *C. coli* isolated varying between 70.5% and 100%. High levels of resistance to ciprofloxacin (52.3%) and tetracyclines (51.5%) were also observed in *C. coli* from fattening pigs, with a lower percentage for erythromycin (15.6%). Combined resistance to ciprofloxacin and erythromycin was reported in 61.2% of *C. coli* isolates from fattening pigs in Spain [40]. In Italy, high rates of ciprofloxacin and tetracycline resistance in *Campylobacter* spp. have been observed and an increasing percentage of *C. coli* strains simultaneously resistant to ciprofloxacin, tetracycline and erythromycin have been found [41]. In our study, we observed that *C. coli* strains of human origin showed a higher resistance than those recovered from pigs and primarily for ciprofloxacin and nalidixic acid (100%); many strains were resistant to erytromicyn (20%). Multi-resistance, defined as resistance to antimicrobials belonging to at least three different classes of antibiotics, was found among 65%, and the predominant R-Type was TETSTRCIPNAL. We have characterized *C. coli* isolated for the presence of resistance genes, and the results of phenotypic and genetic analyses of resistance to tetracycline were fully concordant. All of the strains resistant to tetracycline were shown to carry the gene *tet* (O) responsible for the synthesis of protein Tet(O), which abolishes the inhibitory effect of tetracycline on protein synthesis by a non-covalent modification of the ribosomes [41]. In our study, we noted the presence of the *tet* (O) gene in 88.88% of isolated pigs, indicating a very strong correlation (98.76%) between phenotypic and genotypic resistances, which decrease to the 80% level of concordance in humans. The high resistance to tetracycline of bacteria isolated from food, including *Campylobacter*, remains a serious problem in many European countries [40]. Fluoroquinolone resistance in our strains was mainly due to the presence of the Thr86Ile GyrA mutation. This mutation is the most prevalent in clinical and also in veterinary isolates [35,38]. In our study, it was present in 64.81% and 100% of the isolates from pigs and humans, respectively, displaying a high degree of correlation among the two types of resistances of 96.80% and 100%. Multiple mechanisms associated with antibiotic resistance have been identified in *Campylobacter* spp., but target mutation and drug efflux are most relevant to the resistance to fluoroquinolones and macrolides, as reported in previous studies [42,43]. These two mechanisms function together in conferring a high level of resistance to the two classes of antibiotics. Expression of cmeABC is subject to regulation by CmeR, a repressor encoded by a gene immediately upstream on *cmeA* [43]. CmeABC, a multidrug efflux system in *C. jejuni*, plays an important role in the resistance to different antimicrobials and toxic compounds and it is present also in *C. coli* [44], as reported in other studies [45]. In our study, it was present in all pig and human isolates we analyzed.

## 5. Conclusions

In conclusion, pigs may play a role as an underestimated reservoir of potentially pathogenic *Campylobacter* strains for humans. These animals could contribute to human campylobacteriosis cases and outbreaks through consumption of contaminated meat. The results obtained also provide evidence that antimicrobial resistance is common among *Campylobacter* strains isolated from pigs in Italy, thus indicating the need for continued monitoring and application of reduction strategies within these meat-producing animals. The WHO priority list suggests that the prioritization of research and development of new antibiotics against multidrug-resistant tuberculosis and Gram-negative bacteria is urgently needed. Global research and development strategies should also include antibiotics active against more common community bacteria, such as antibiotic-resistant *Salmonella* spp, *Campylobacter* spp, and *Helicobacter pylori* [42].

## Figures and Tables

**Figure 1 microorganisms-08-00222-f001:**
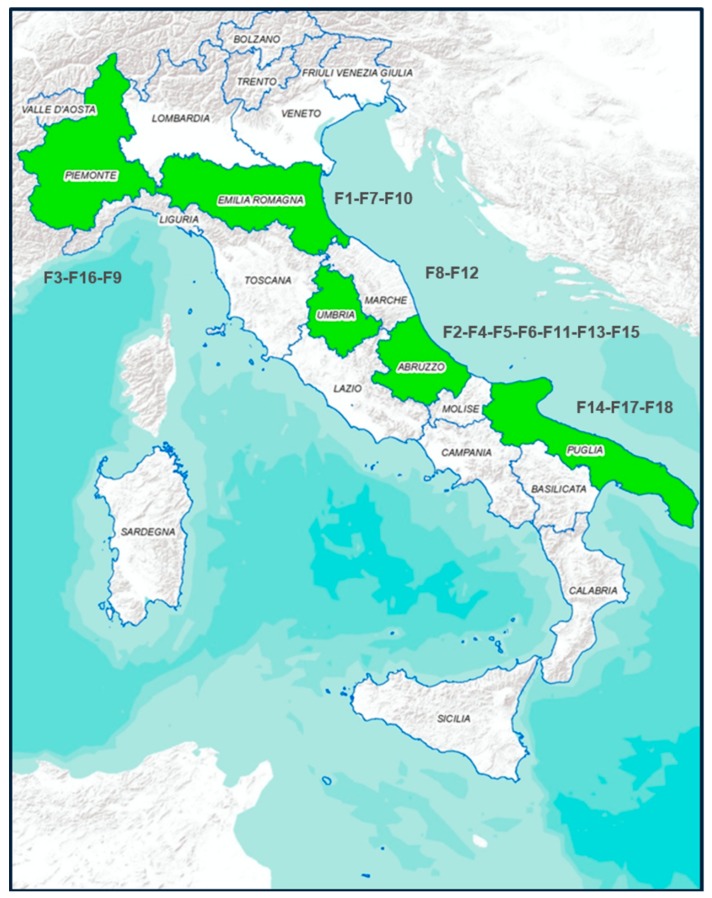
In green are the Italian Regions where pigs were bred.

**Figure 2 microorganisms-08-00222-f002:**
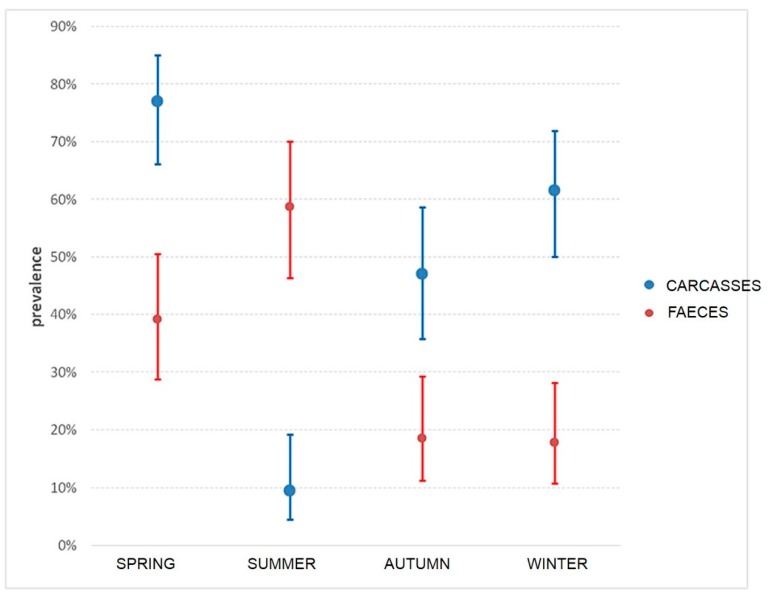
Prevalence and CI in carcasses and faeces during seasons.

**Figure 3 microorganisms-08-00222-f003:**
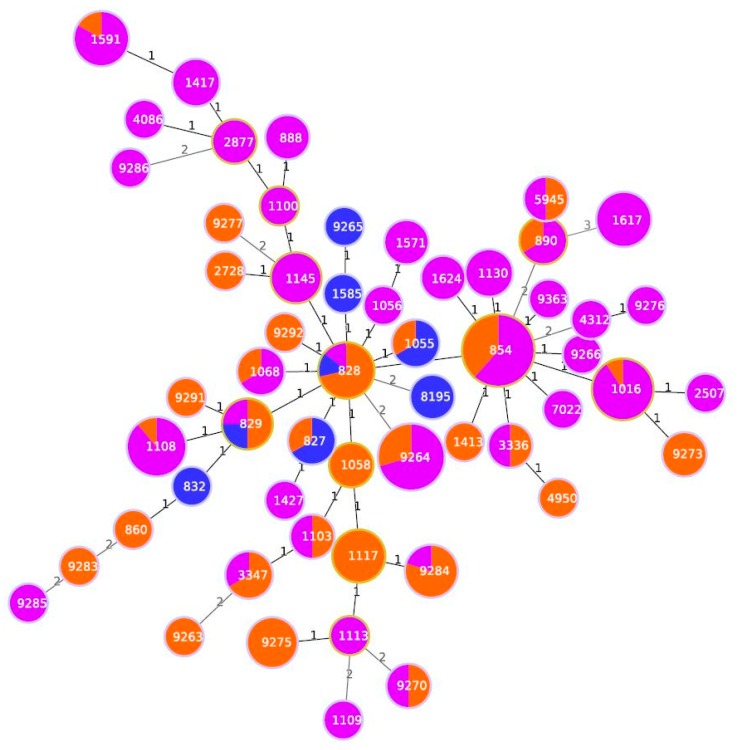
Minimum spanning tree generated for 11 human isolates and 162 pig isolates. Each circle represents an allelic profile. The numbers on the connecting lines illustrate the numbers of target genes with differing alleles. The different sources are distinguished by the colors of the circles with blue for human isolates, orange for pig stool and violet for pig carcasses isolates.

**Figure 4 microorganisms-08-00222-f004:**
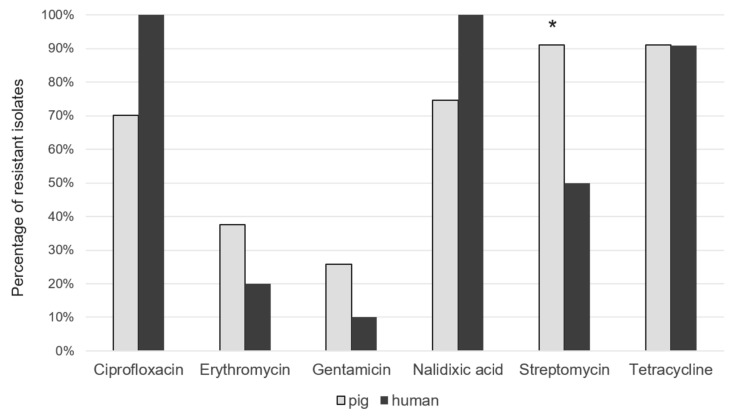
Percentage of resistant *C. coli* isolates from pig and humans. * *p* ≤ 0.5.

**Table 1 microorganisms-08-00222-t001:** The most representative Pulsed-field gel electrophoresis (PFGE) pulsotypes and multi-locus sequence typing (MLST) profiles from *Campylobacter coli* isolated from swine per farm.

PFGE	MLST Profile	FARM 1	FARM 2	FARM 3	FARM 4	FARM 5	FARM 6	FARM 7	FARM 9	FARM 10
**P1**	ST-1617	Winter (C)	Autumn, Spring (C)	/	/	/	Spring (C)		/	/
	ST-9264	Winter (C)	Winter (F)(C), Spring (C)	Spring (C)	/	/	/	Spring (F)(C)	/	/
	ST-1624 *	/	Spring (C)	/	/	/	/	/	/	/
**P2**	ST-1016 *	/	Autumn (C)	Spring (C)	Autumn (C), Spring (C)	Autumn(C)(F)	Winter (C), Spring (C)	/	/	/
**P3**	ST-1591	/	Spring (C)(F)	/	/	/	/	/	/	/
	ST-1417	/	Spring (C)	/	/	/	/	/	/	/
	ST-9277	/	/	/	/	/	Spring (F)	/	/	/
	ST-1113 *	/	/	Spring (C)	/	/	/	/	/	/
**P4**	ST-854 *	Winter (C)	Spring (C)(F)	/	/	/	Spring (C)(F) **	/	/	/
	ST-9276	/	/	/	/	/	Spring (C)	/	/	/
	ST-9275	/	Summer (F)	/	/	/	/	/	/	/
	ST-1130 *	/	/	Spring (C)	/	/	/	/	/	/
**P5**	ST-1117 *	/	Spring (F)	/	/	/	/	/	/	/
	ST-9264	/	Winter (F), Spring (C)	/	/	/	/	Spring (F)(C)	/	/
	ST-9284	/	/	/	/	/	/	Summer (F)	/	/
**P6**	ST-9291	/	/	/	/	/	/	/	Summer (F) **	
	ST-854 *	/	Winter (C), Spring (F) **	/	/	/	Spring (C)(F) **	/	/	/
	ST-828 *	/	/	/	/	/	Winter (C), Spring (F) **	/	/	/
**P7**	ST-1427 *	Winter (C)	/	/	/	/		/	/	/
	ST-1108 *	Winter (C)	/	/	/	/	Summer (F), ** Winter (C) ***	/	/	Winter (C)

*= 828 complex, C= carcasses; F= faeces, **= HU provenance, ***= F provenance

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
