# Peer review of "Prevalence, Population Diversity and Antimicrobial Resistance of Campylobacter coli Isolated in Italian Swine at Slaughterhouse"

_microorganisms, 2020, doi:10.3390/microorganisms8020222_

Round 1
Reviewer 1 Report
This manuscript presented scientific information about C. coli spp. isolated from feces and carcasses of pigs slaughtered in Italy. The authors have presented data about the prevalence of C. coli, phenotypic and genotypic antimicrobial resistance, and genetic similarity of C. coli isolated from human and pigs. Overall the manuscript is written with easy to understand language, however several of the expression require revisions.
Abstract. Line 24, remove brackets,
Line 26-27, revise the sentence “The genetic detection determinants retrieved five resistance genes”. Five resistance genes were detected along with mutation in gyrA gene.
Introduction: line 38, change the express, to a minor extend by direct contact, what does this mean,
Line 44, change to “are responsible for about….
Line 47, revise to “C. coli has been found to be responsible for about …. “
M&M; How many carcass samples and how many fecal samples were collected, is the same pigs was sampled for both feces and swab sample.
Is the skin was removed from the carcasses or only scalding was done. Or carcasses received any wash for decontamination.
Line 63 revise the expression “sessions”, it can be 16 trips to the slaughterhouse. With four trips each season.
Line 64, revise “during slaughter session” to “a total of 12,308 animals were slaughtered”.
Line 66 revise “session” to visit.
line 68 fattening phase of pigs was carried out…
line 70 uniquely identified what does this mean.
Line 71, revise : while swab samples from carcasses were collected before cooling in….
Line 72 what are the withdrawal points.
Line 76, 77, change to microaerophilic
Line 80 “spp. were then submitted ….”
Line 123-124, use three lettered abbreviations for antibiotics
Line 129, why sued an old CLSI criteria.
Results: Line 140-141, what is the value of this information.
Line 143 add word carcass before swab
Line 146, revise “prevalence of C. coli was higher….”
Line 147, revise to “Camp. Contamination was low on carcasses”
Line 150, use instead “with both typing methods”,
Line 151, resulted in 25 clonal population, is this mean 25 clones, or 25 genetically similar patterns.
Line 156, ST-828
Line 167, antimicrobial resistance of C. coli
Line 173, resistome, revise it
Conclusions is presented in a logical manner in support of data presented.
The figures and tables are appropriate.
References are appropriate.
Author Response
Open Review
(x) I would not like to sign my review report
( ) I would like to sign my review report
English language and style
(x) Extensive editing of English language and style required
( ) Moderate English changes required
( ) English language and style are fine/minor spell check required
( ) I don't feel qualified to judge about the English language and style
|
Yes |
Can be improved |
Must be improved |
Not applicable |
|
|
Does the introduction provide sufficient background and include all relevant references? |
(x) |
( ) |
( ) |
( ) |
|
Is the research design appropriate? |
( ) |
( ) |
(x) |
( ) |
|
Are the methods adequately described? |
( ) |
( ) |
(x) |
( ) |
|
Are the results clearly presented? |
( ) |
(x) |
( ) |
( ) |
|
Are the conclusions supported by the results? |
(x) |
( ) |
( ) |
( ) |
Comments and Suggestions for Authors
This manuscript presented scientific information about C. coli spp. isolated from feces and carcasses of pigs slaughtered in Italy. The authors have presented data about the prevalence of C. coli, phenotypic and genotypic antimicrobial resistance, and genetic similarity of C. coli isolated from human and pigs. Overall the manuscript is written with easy to understand language, however several of the expression require revisions.
Abstract. Line 24, remove brackets,
Changed accordingly
Line 26-27, revise the sentence “The genetic detection determinants retrieved five resistance genes”. Five resistance genes were detected along with mutation in gyrA gene.
Modified as suggested from the reviewer
Introduction: line 38, change the express, to a minor extend by direct contact, what does this mean,
Amended according to the comment. The sentence has been changed with “…and less frequently by direct contact with…”
Line 44, change to “are responsible for about….
Modified as suggested from the reviewer
Line 47, revise to “C. coli has been found to be responsible for about …. “
Modified as suggested from the reviewer
M&M; How many carcass samples and how many fecal samples were collected, is the same pigs was sampled for both feces and swab sample.
A total of 280 pig carcasses were sampled at the slaughterhouse.
You can read in lines 64-65: “A total of 12,308 animals were slaughtered, among which 280 animals were randomly selected (1 carcass for every 40-50 animals, about 17-18 animals sampled for each visit)”…...and in lines 70-72: “From each animal, the fecal content was taken immediately after the evisceration phase while swab samples from carcasses were collected before cooling…”
Is the skin was removed from the carcasses or only scalding was done. Or carcasses received any wash for decontamination.
The skin was not removed from the carcasses and the carcasses have not received any wash for decontamination.
Line 63 revise the expression “sessions”, it can be 16 trips to the slaughterhouse. With four trips each season.
Changed accordingly.
Line 64, revise “during slaughter session” to “a total of 12,308 animals were slaughtered”.
Changed accordingly
Line 66 revise “session” to visit.
Changed accordingly
line 68 fattening phase of pigs was carried out…
Changed accordingly
line 70 uniquely identified what does this mean.
We thank the reviewer for this comment. The sentence has been rewritten as follow: “From each animal, the fecal content was taken immediately after…”
Line 71, revise: while swab samples from carcasses were collected before cooling in….
Changed accordingly
Line 72 what are the withdrawal points.
The withdrawl point have been changed in the manuscripty in: ham, back, belly, jowl as reported by the EU law 2001/471/CE.
Line 76, 77, change to microaerophilic
Changed accordingly
Line 80 “spp. were then submitted ….”
Changed accordingly
Line 123-124, use three lettered abbreviations for antibiotics
Changed accordingly
Line 129, why sued an old CLSI criteria.
Thank you for this observations. We changed the sentence as follow: “Susceptibility to antimicrobials was evaluated with the microdilution method using the Sensititre automated system (TREK Diagnostic Systems, Italy) following the harmonised rules for the monitoring and reporting of AMR in Europe (Commission Implementing Decision 2013/652/EC). Colonies were cultured on Columbia agar for 48 hours in microaerophilic atmosphere, inoculated in Mueller Hinton Broth supplemented with blood and dispensed into Eucamp microtiter plates (TREK Diagnostic Systems, Biomedical Service, Italy), containing known scalar concentrations of the following antimicrobial substances: gentamicin (GEN) (0.12 - 16 µg/ml), streptomycin (STR) (1 - 16 µg/ml), ciprofloxacin (CIP) (0.06 - 4 µg/ml), tetracycline (TET) (0.25 - 16 µg/ml), erythromycin (ERY) (0.5 - 32 µg/ml), nalidixic acid (NAL) (2 - 64 µg/ml), and chloramphenicol (CHL) (2 - 32 µg/ml). The plates were then incubated at 42°C in microaerobic atmosphere for 24 hours. To evaluate the MICs of the isolates, Swin v3.3 Software (Thermo Fisher Scientific) was used in accordance with the epidemiological cutoff values (ECOFFs) as defined by EUCAST (European Committee on antimicrobial breakpoints) (www.eucast.org) to interpret their antimicrobial susceptibilities. C. jejuni strain NCTC 11351 was included for the quality control of the minimal inhibitory concentration (MIC) test”.
“
Results: Line 140-141, what is the value of this information.
This sentence has been rewritten with more informations.
Line 143 add word carcass before swab
Changed accordingly
Line 146, revise “prevalence of C. coli was higher….”
Changed accordingly
Line 147, revise to “Camp. Contamination was low on carcasses”
Changed accordingly
Line 150, use instead “with both typing methods”,
Changed accordingly
Line 151, resulted in 25 clonal population, is this mean 25 clones, or 25 genetically similar patterns.
The authors refer to 25 genetically similar patterns.
Line 156, ST-828
Changed accordingly
Line 167, antimicrobial resistance of C. coli
Changed accordingly
Line 173, resistome, revise it
We thank to reviewer for this comment. The sentence has been rewritten as follow: “The analysis of the isolates allowed to identify several resistance genes that..”
Conclusions is presented in a logical manner in support of data presented.
The figures and tables are appropriate.
References are appropriate.
Submission Date
03 January 2020
Date of this review
16 Jan 2020 23:02:36

Reviewer 2 Report
This paper described the “Prevalence, population diversity and antimicrobial resistance of Campylobacter coli isolated in Italian swine at slaughterhouse”.
This paper is well written and interesting.
However, major revisions should be made. You have a lot of interesting data but not always valued. Some data and some explanations of the data are missing, and some data could be presented differently to better understand the results that you obtain.
Here my comments:
First, in all the text, all the cited Figures and tables have to be cited inside brackets as for example (Figure 1), (Table 1) etc..
Abstract :
Sentence on prevalence is missing. I suggest you to add line 21 this sentence “…….isolated from humans. The prevalence of contamination was higher on carcasses (50.4%) than in faeces (32.9%). The 162 C. coli …”. If the abstract too long after that, you suggest you to remove the first sentence which is not necessary.
Line 28 : remove this sentence “This study shows prevalence, phylogenetic diversity and AMR profiles of C. coli in swine.” Not necessary.
Introduction :
Line 38 : you cited the reference 2 for “….of contaminated water or food [2]…..” or this reference is from Thakur and on swine. You must cite another reference.
Line 49-50 : move the reference [6] in this sentence as suggested, indeed, it is not well placed and add reference for pig as suggested “ C. jejuni is considered prevalent in poultry [5] and cattle [6] while pigs are mostly implicated as reservoirs of C. coli [Kempf et al., 2017]
Materials and methods :
Line 68 : add the number of farms concerned by this study “….the fattening phase was carried out in Italy, in X farms (coded F1 to Fx) located in….. “
and on the figure 1, place the farm F, F2, F3, etc to visualize in which regions where are the farms, farms from which come the sampled pigs.
Line 79 : correct “…. according respectively to the part 1 and to the part 2 of the EN ISO 10272-2006 on swab samples.”
Line 79 : how many isolates did you identify for the species by PCR?
in results you write line 152 : a total of 162 C. coli was isolated from pig. Then 162 isolates were typed for PFGE and MLST. Why in table 2 for antibiotic resistance, when I count the number of isolates per antibiotic, it is 221 isolates?
Line 91 : Salmonella in italic
Line 110 : review the sentence “….. were identified using the MLST database available at [15]. …” word is missing? Or do you mean “….. were identified using the MLST database available online [15]. …”
Line 129 : I suggest you to add this sentence “ Strains were considered resistant when MIC break points were ≥ to 4, ≥ to 8, ≥ to 16, ………, for respectively Ciprofloxaxin, Erytromycin , …
Line 129 : start this paragraph on the next line “ C. coli genome assemblies were searched for genomic…..” . I do not understand. Have you sequenced the genome of your strains? I imagine so if you want to see the sequences of the ATB resistance genes. In this case, why it is not indicated and the method presented?
Line 137 : in results, you give data on Gyr A and mutation position 86, but also on Cme genes, and on Tet(O). Why it is not indicated in method, that you are looking for the presence of these genes? Moreover, did you look for other AMR genes or only for these genes and why?
Results :
first, put in italic Campylobacter and C.coli in all the text
Line 139 : How many pig per farm were sampled finally? Complete the sentence “ A total of 280 pig carcasses were sampled at the slaughterhouse coming from different part of Italian Regions (Figure 1); the number of pigs finally sampled per farm was x, x, x, ….. for farm F1, F2, F3, ……respectively. Among the 280 pigs, 162 were females (10.7%) and 118 males (89.3%) ….”
line 150 : start this paragraph on the next line “ A high genetic diversity for C. coli was observed, whatever the typing method”.
Line 152 move the sentence “A total of 162 C. coli was isolated from swabs and faeces samples collected on pigs at slaughtering” be before line 150 “A high genetic diversity for C. coli was observed…”
Line 156 : write ST-828 clonal complex instead of 828ST-156 clonal complex.
All this following paragraph has to be reviewed. It is not clear for me in which farm are the ST, table 1 does not allow comparison with the data written in the text. Also, it is confusing because you give data on ST and then on genotypes in the text, and it is very difficult to understand the distribution of the ST in the farm, in the countries, and what you are talking about ST or PFGE, genotypes?. How many genotypes coming from each Farm?
“Among the pig isolates, the most frequent STs were ST854 (25.74%), ST9264 (16.83%) and ST1016 (11.88%) Figure 3. Sequence types ST828 (6.93%) and ST829 (3.96%) were present both in human and pig isolates Figure 3. ST854 clone was common to all farms located in different regions; ST828 clone turned out to be the only clone circulating in all four seasons. In many animals, the same clone was isolated in both the faeces and the carcass; the comparison of the STs by country of origin of the animals showed the presence of the same STs in the majority of cases, although distinct clones between animals born in Italy and abroad were found in 27.6% of the pigs. Each analyzed farm was characterized by different clonal populations, composed from 1 to 5 different genotypes, with the exception of farm 2 with 11 circulating genotypes. Table 1. Some genotypes such as ST9264, ST9276 and ST9284 were isolated in subsequent seasons in animals coming from the same farms Table 1.”
For example :
Line 158 : you write “Sequence types ST828 (6.93%) and ST829 (3.96%) were present both in human and pig isolates” or in figure 3, I see also ST1055 and ST827 both in pig and human. Why they are not cited?.
Line 159 : you write “ST854 clone was common to all farms located in different regions” or in table 1, I see this ST854 only in farm 1, 2 and 6? Specify in which farm is this ST
Line 160 : you write “ST828 clone turned out to be the only clone circulating in all four seasons”. Or I see only it for Farm 6 in spring (c) in the table 1. Are you speaking about the ST828 or about the ST-828 clonal complex?
Etc….
Table 1 must be presented differently because it is difficult to found the data. In this table write Stxxx instead xxxST, and I suggest you to put a * as for example ST1427* instead of writing 1427ST-828 complex to facilitate reading. It was with this table that I assumed that there were 9 farms. Is it true?
At each ST, it will be good to put the number of isolates in parenthesis concerned by the ST , for exp : ST1427 Winter (5) (C)
line 170 : add this sentence “ Only 37.55% and 23.98% of the strains from pigs were resistant to Erytromycin and Gentamycin, repectively”. is there significant difference between pigs and humans?
table 2/3: I suggest you to replace them in one figure like histogram with for each antibiotic a column indicating the % of Res to ATB for pork and next to it another column indicatig the % of Res of ATB for human. It’s enough.
Line 173 : you write “The analysis of the resistoma of the isolates allowed to identify several resistance genes that include: gyrA, tet (O), cmeA, cmeB, cmeC and cmeR, (Figure 1) ” How many genes in total? Is the figure 4 necessary because you give the data in the text. I suggest you to remove figure 4.
Discussion
Line 217-209 : you write “The higher prevalence of Campylobacter in carcasses compared to faeces could indicate a cross contamination during slaughter process, argued by the low level of contamination and also confirmed by PFGE analysis.” I did not read in the results data which explains the cross contamination on the basis of the PFGE profiles. This data should be explained in the Result section of this paper.
Line 221 : you cite a reference on poultry [29 ]while there is data on pigs for effect of warm season on campylobacter prevalence. Cite for exp : Denis et al., 2011, Campylobacter from sows in farrow-to-finish pig farms: risk indicators and genetic diversity. Vet microbiol. who showed warmer months as risk factor.
Line 231 : you write “Each farm from which the tested pigs originated resulted characterized ……………by different clonal populations, belonging to 1-11 different genotypes”. Like previously, when you speak about genotypes, I do not know if it is PFGE, and the distribution of these genotypes in each farm is missing in results.
Reference :
Add suggested reference
Harmonize references; they are not all written in the same format.
Author Response
Open Review
(x) I would not like to sign my review report
( ) I would like to sign my review report
English language and style
( ) Extensive editing of English language and style required
( ) Moderate English changes required
(x) English language and style are fine/minor spell check required
( ) I don't feel qualified to judge about the English language and style
Yes Can be improved Must be improved Not applicable
Does the introduction provide sufficient background and include all relevant references?
(x) ( ) ( ) ( )
Is the research design appropriate?
(x) ( ) ( ) ( )
Are the methods adequately described?
( ) ( ) (x) ( )
Are the results clearly presented?
( ) ( ) (x) ( )
Are the conclusions supported by the results?
(x) ( ) ( ) ( )
Comments and Suggestions for Authors
This paper described the “Prevalence, population diversity and antimicrobial resistance of Campylobacter coli isolated in Italian swine at slaughterhouse”.
This paper is well written and interesting.
However, major revisions should be made. You have a lot of interesting data but not always valued. Some data and some explanations of the data are missing, and some data could be presented differently to better understand the results that you obtain.
Here my comments:
First, in all the text, all the cited Figures and tables have to be cited inside brackets as for example (Figure 1), (Table 1) etc..
Abstract :
Sentence on prevalence is missing. I suggest you to add line 21 this sentence “…….isolated from humans. The prevalence of contamination was higher on carcasses (50.4%) than in faeces (32.9%). The 162 C. coli …”. If the abstract too long after that, you suggest you to remove the first sentence which is not necessary.
We thank the reviewer for this comment. Amended according to the comment.
Line 28 : remove this sentence “This study shows prevalence, phylogenetic diversity and AMR profiles of C. coli in swine.” Not necessary.
Amended according to the comment.
Introduction :
Line 38 : you cited the reference 2 for “….of contaminated water or food [2]…..” or this reference is from Thakur and on swine. You must cite another reference.
Changed accordingly. The correct reference is now reported in the Reference section.
Line 49-50: move the reference [6] in this sentence as suggested, indeed, it is not well placed and add reference for pig as suggested “ C. jejuni is considered prevalent in poultry [5] and cattle [6] while pigs are mostly implicated as reservoirs of C. coli [Kempf et al., 2017]
Amended according to the comment.
Materials and methods :
Line 68 : add the number of farms concerned by this study “….the fattening phase was carried out in Italy, in X farms (coded F1 to Fx) located in….. “
Changed accordingly. The sentence has been rewritten as follow: “For all the animals sampled, the fattening phase of pigs was carried out in Italy, in 18 farms (coded F1 to F18) located in different regions of north (Piemonte, F3, F16, F9 and Emilia Romagna F1, F7, F10), central (Umbria F8, F12 and Abruzzo F2, F4, F5, F6, F11, F13, F15) and south Italy (Puglia F14, F17, F18) Figure 1”.
and on the figure 1, place the farm F, F2, F3, etc to visualize in which regions where are the farms, farms from which come the sampled pigs.
Amended according to the comment.
Line 79 : correct “…. according respectively to the part 1 and to the part 2 of the EN ISO 10272-2006 on swab samples.”
Amended according to the comment.
Line 79 : how many isolates did you identify for the species by PCR?
We thank the reviewer for this comment. This information is reported in Result section. In detail, “…A total of 162 C. coli was isolated from swabs and faeces samples collected on pigs at slaughtering.”(lines 153-154).
in results you write line 152 : a total of 162 C. coli was isolated from pig. Then 162 isolates were typed for PFGE and MLST. Why in table 2 for antibiotic resistance, when I count the number of isolates per antibiotic, it is 221 isolates?
We thank the reviewer for this observation. The explanation of the different number of strains analysed is because we had available 162 assemblies in the NRL for Campylobacter, therefore for a better comparison we decided to use the same number of strains for all typing methods.
We reformulated the sentence as follow: “A total of 221 C. coli was isolated from swabs and faeces samples collected on pigs at slaughtering. A high genetic diversity for C. coli was observed, with both typing method. PFGE profiles of C. coli strains, after SmaI and KpnI enzyme digestion, resulted in 25 clonal populations, according to a similarity of 90%. The most representative (P1-P7) are shown in Table 1.”
Line 91 : Salmonella in italic
Amended according to the comment.
Line 110 : review the sentence “….. were identified using the MLST database available at [15]. …” word is missing? Or do you mean “….. were identified using the MLST database available online [15]. …”
Amended according to the comment.
Line 129 : I suggest you to add this sentence “ Strains were considered resistant when MIC break points were ≥ to 4, ≥ to 8, ≥ to 16, ………, for respectively Ciprofloxaxin, Erytromycin , …
Thanks to reviewer for this suggestion. Changed accordingly.
Line 129 : start this paragraph on the next line “ C. coli genome assemblies were searched for genomic…..” . I do not understand. Have you sequenced the genome of your strains? I imagine so if you want to see the sequences of the ATB resistance genes. In this case, why it is not indicated and the method presented?
We thank the reviewer for this observation. The assemblies that we used in this work were available at the NRL for Campylobacter; therefore, we didn’t include the method in our article. So, the sentence has been reformulated as follow: “C. coli genome assemblies, available at the NRL for Campylobacter, were searched for genomic AMR traits presence.” We can supply the fasta sequences, as supplementary material, if the reviewer request it.
Line 137 : in results, you give data on Gyr A and mutation position 86, but also on Cme genes, and on Tet(O). Why it is not indicated in method, that you are looking for the presence of these genes? Moreover, did you look for other AMR genes or only for these genes and why?
We thank the reviewer for this comments and observations. We reported in lines 132-136 the part related to AMR genes and method. We looked these genes because are the first-line antibiotics used for the campylobacteriosis treatment in humans and one of the aims of this work was to do also a comparison between the two types of antimicrobial resistances.
Results :
first, put in italic Campylobacter and C.coli in all the text
Changed accordingly.
Line 139 : How many pig per farm were sampled finally? Complete the sentence “ A total of 280 pig carcasses were sampled at the slaughterhouse coming from different part of Italian Regions (Figure 1); the number of pigs finally sampled per farm was x, x, x, ….. for farm F1, F2, F3, ……respectively. Among the 280 pigs, 162 were females (10.7%) and 118 males (89.3%) ….”
Amended according to the comment.
line 150 : start this paragraph on the next line “ A high genetic diversity for C. coli was observed, whatever the typing method”.
It has been already changed with “…,with both typing method”, as suggested by another reviewer.
Line 152 move the sentence “A total of 162 C. coli was isolated from swabs and faeces samples collected on pigs at slaughtering” be before line 150 “A high genetic diversity for C. coli was observed…”
Changed accordingly.
Line 156 : write ST-828 clonal complex instead of 828ST-156 clonal complex.
Changed accordingly.
All this following paragraph has to be reviewed. It is not clear for me in which farm are the ST, table 1 does not allow comparison with the data written in the text. Also, it is confusing because you give data on ST and then on genotypes in the text, and it is very difficult to understand the distribution of the ST in the farm, in the countries, and what you are talking about ST or PFGE, genotypes?. How many genotypes coming from each Farm?
“Among the pig isolates, the most frequent STs were ST854 (25.74%), ST9264 (16.83%) and ST1016 (11.88%) Figure 3. Sequence types ST828 (6.93%) and ST829 (3.96%) were present both in human and pig isolates Figure 3. ST854 clone was common to all farms located in different regions; ST828 clone turned out to be the only clone circulating in all four seasons. In many animals, the same clone was isolated in both the faeces and the carcass; the comparison of the STs by country of origin of the animals showed the presence of the same STs in the majority of cases, although distinct clones between animals born in Italy and abroad were found in 27.6% of the pigs. Each analyzed farm was characterized by different clonal populations, composed from 1 to 5 different genotypes, with the exception of farm 2 with 11 circulating genotypes. Table 1. Some genotypes such as ST9264, ST9276 and ST9284 were isolated in subsequent seasons in animals coming from the same farms Table 1.”
For example :
Line 158 : you write “Sequence types ST828 (6.93%) and ST829 (3.96%) were present both in human and pig isolates” or in figure 3, I see also ST1055 and ST827 both in pig and human. Why they are not cited?.
We reformulated the sentence.
Line 159 : you write “ST854 clone was common to all farms located in different regions” or in table 1, I see this ST854 only in farm 1, 2 and 6? Specify in which farm is this ST
We reformulated the sentence and the table1.
Line 160 : you write “ST828 clone turned out to be the only clone circulating in all four seasons”. Or I see only it for Farm 6 in spring (c) in the table 1. Are you speaking about the ST828 or about the ST-828 clonal complex?
We reformulated the sentence and the table 1.
Etc….
Table 1 must be presented differently because it is difficult to found the data. In this table write Stxxx instead xxxST, and I suggest you to put a * as for example ST1427* instead of writing 1427ST-828 complex to facilitate reading. It was with this table that I assumed that there were 9 farms. Is it true?
Changed accordingly. We presented a new table 1 with more information, including also the farms, the seasons, the provenance of the pig and the matrices.
At each ST, it will be good to put the number of isolates in parenthesis concerned by the ST , for exp : ST1427 Winter (5) (C)
Amended according to the comment.
Regarding the sentences from lines 159 to 167 (Result section), now you can read:
“Among the pig isolates, the most frequent STs were ST-854 (16.04%), ST-9264 (10.49%) ST-1016 (6.8%) and ST-1108 (5.55%) Figure 3. Sequence types ST-828 (4.05%), ST-829 (2.31%), ST-1055 and ST-827 (1.73%) were present both in human and pig isolates Figure 3. Each analyzed farm was characterized by different STs and nine of them by seven different PFGE pulsotypes (P1-P7). Table 1. In particular, farm 2, showed the presence of 10 different STs followed by farm 6 with 7 STs. Table 1. It is apparent that some ST profiles of C. coli appears to dominate in a geographic area for a variable period. For example, some genotypes such as ST-1617, ST-9264, ST-1016, ST-828 and ST-1108 were isolated in subsequent seasons in animals coming from the same farms Table 1. On the other hand, the same ST was also found on different farms and, with the exception of two of these cases (ST-9264 and 854-ST) were the isolates had the same PFGE pulsotype. This is the case of ST-1016 predominant in 5 farms (F2-F3-F4-F5-F6) with PFGE pulsotype P2; ST-1617 and ST-1108 respectively common in 3 farms (F1, F2, F6) and (F2, F6 and F10) with PFGE pulsotypes P1 and P7. Table 1. ST-9264 and ST-854 were found to be the most diverse ST with 2 different PFGE types. In many animals, the same ST was isolated in both the faeces and the carcass. Table 1. The comparison of the STs by country of origin of the animals showed the presence of the same STs in the majority of cases, although distinct STs between animals born in Italy and abroad were found in 27.6% of the pigs”.
line 170 : add this sentence “ Only 37.55% and 23.98% of the strains from pigs were resistant to Erytromycin and Gentamycin, repectively”. is there significant difference between pigs and humans?
Changed accordingly.
table 2/3: I suggest you to replace them in one figure like histogram with for each antibiotic a column indicating the % of Res to ATB for pork and next to it another column indicatig the % of Res of ATB for human. It’s enough.
Line 173 : you write “The analysis of the resistoma of the isolates allowed to identify several resistance genes that include: gyrA, tet (O), cmeA, cmeB, cmeC and cmeR, (Figure 1) ” How many genes in total? Is the figure 4 necessary because you give the data in the text. I suggest you to remove figure 4.
The total genes analyzed were 6. The figure has been removed.
Discussion
Line 217-209 : you write “The higher prevalence of Campylobacter in carcasses compared to faeces could indicate a cross contamination during slaughter process, argued by the low level of contamination and also confirmed by PFGE analysis.” I did not read in the results data which explains the cross contamination on the basis of the PFGE profiles. This data should be explained in the Result section of this paper.
The discussion has been rewritten and the Table 1 now includes more informations.
Line 221 : you cite a reference on poultry [29 ]while there is data on pigs for effect of warm season on campylobacter prevalence. Cite for exp : Denis et al., 2011, Campylobacter from sows in farrow-to-finish pig farms: risk indicators and genetic diversity. Vet microbiol. who showed warmer months as risk factor.
Thank you for this observation. The reference was added.
Line 231 : you write “Each farm from which the tested pigs originated resulted characterized ……………by different clonal populations, belonging to 1-11 different genotypes”. Like previously, when you speak about genotypes, I do not know if it is PFGE, and the distribution of these genotypes in each farm is missing in results.
This part of the discussion has been rewritten. Now the reviewer can read: Each farm from which the tested pigs originated resulted characterized by different clonal populations, characterized by several STs.
Reference :
Add suggested reference
It was done.
Harmonize references; they are not all written in the same format.
It was done.
Submission Date
03 January 2020
Date of this review
17 Jan 2020 14:25:28

Round 2
Reviewer 2 Report
I thank the authors for responding to my requests. The data from this study are much better explained in the text and better presented in the figures and table. It is now much clearer.some errors remain: line 66 : cite ref [7] and not [Kempf et al., 2017] line 212, 217, 220 : put in italic Campylobacter, C. coli, C. jejuni line 243 : start paragraph "the highest level …." on a new line. line 425 : you have changed the ref [30] now on pigs but in the text you have let chicken. Replace the word "chicken" by "pig" line 742 : start the ref [29] on a new line
Author Response
line 66 : cite ref [7] and not [Kempf et al., 2017]
Changed accordingly
line 212, 217, 220 : put in italic Campylobacter, C. coli, C. jejuni
Changed accordingly
line 243 : start paragraph "the highest level …." on a new line.
Changed accordingly
line 425 : you have changed the ref [30] now on pigs but in the text you have let chicken. Replace the word "chicken" by "pig"
Changed accordingly
line 742 : start the ref [29] on a new line
Changed accordingly.